# *Ficus benghalensis* extract mediated green synthesis of silver nanoparticles, its optimization, characterization, computational studies, and its *in vitro* and *in vivo* biological potential

Inam Ud Din[1], Rahaf Ajaj[2]*, Abdur Rauf [3]*, Zubair Ahmad[3], Naveed Muhammad[4], Shahid Ali[1], Hassan A. Hemeg[5], Imran Ullah [6]

**1** Materials Research Laboratory, Department of Physics, University of Peshawar, Peshawar, Pakistan, **2** Department of Environmental and Public Health, College of Health Sciences, Abu Dhabi University, Abu Dhabi, United Arab Emirates, **3** Department of Chemistry, University of Swabi, Anbar, Khyber Pakhtunkhwa, Pakistan, **4** Department of Pharmacy, Abdul Wali Khan University, Khyber Pakhtunkhwa, Pakistan, **5** Department of Medical Laboratory Technology, College of Applied Medical Sciences, Taibah University, Al-Medinah Al-Monawara, Saudi Arabia, **6** Institute for Crystallography and Structural Physics, Friedrich-Alexander University, Erlangen-Nurnberg, Germany

* abdurrauf@uoswabi.edu.pk (AR); rahaf.ajaj@adu.ac.ae (RA)

## Abstract

In this work, Silver (Ag) nanoparticles (NPs) were synthesized via green synthesis using *Ficus benghalensis* root extract (FBRE), serving as a capping and stabilizing agent. The synthesized Ag NPs were characterized via complementary characterization techniques, including SEM, XRD, EDS, UV-Vis, and FT-IR. SEM analysis revealed the fabrication of spherical NPs with an average size of 41.55 nm. A plasmon resonance peak was observed at 430 nm. FBRE effectively capped and stabilized the Ag NPs, ensuring their structural integrity over time, and is confirmed via FT-IR scan. DFT calculation revealed a thermodynamically and mechanically stable system. Moreover, optoelectronic properties confirmed the metallic behavior of Ag with a major contribution from 4d orbital near the fermi level and 5s orbital contribution to the conduction band with light absorption in the visible spectrum. Biological evaluations demonstrated significant enzyme inhibition. Ag NPs inhibited urease (80.76%), α-glucosidase (80.98%), carbonic anhydrase II (89.32%), and xanthine oxidase (49.9%), outperforming FBRE. *In Vivo*, Ag NPs exhibited dose-dependent analgesic (83.09% writhing inhibition at 10 mg/kg, similar to diclofenac) and sedative (16.09% locomotor reduction at 10 mg/kg) effects. Molecular docking confirmed strong enzyme-ligand interactions. These findings highlight the biomedical potential of FBRE-synthesized Ag NPs, particularly for enzyme inhibition and pharmacological applications.

**Data availability statement:** All relevant data are contained within the paper.

**Funding:** The author(s) received no specific funding for this work.

**Competing interests:** NA.

## 1. Introduction

Metallic nanoparticles (NPs) have remained a prominent field of study for decades, and they have unique properties due to their higher surface-to-volume ratio compared to bulk materials. Versatile properties of NPs have extensive applications in optics, electronics, and medicine, especially in drug delivery systems [1–4]. NPs have been extensively studied for various applications so far. Among metallic NPs, noble metal NPs, including gold (Au) and Ag NPs, are highly regarded in nanoscience due to their biocompatibility and minimal toxicity to human health. Ag NPs can be synthesized through various routes, including chemical, physical, and green synthesis. Post synthesis, the key concerns are biocompatibility, toxicity, and an environmentally friendly synthesis approach. Green synthesis approaches for NPs synthesis have gained attention as sustainable alternatives that involve enzymes, microorganisms, and plant extracts (roots, stems, leaves, etc.) [5]. Green synthesis of AgNPs has emerged as a sustainable and eco-friendly alternative to conventional chemical and physical synthesis methods. Traditional approaches often rely on toxic reducing agents such as sodium borohydride and organic solvents, which pose environmental hazards and limit biomedical applications due to potential cytotoxicity. In contrast, green synthesis utilizes plant extracts, microorganisms, or biomolecules as reducing and stabilizing agents, offering a cost-effective, non-toxic, and biodegradable route. Additionally, the bioactive compounds present in plant extracts not only aid in nanoparticle stabilization but also enhance their therapeutic potential, making them suitable for biomedical and pharmaceutical applications. Compared to high-energy-consuming physical methods like laser ablation or microwave-assisted synthesis, green synthesis is energy-efficient and operates under ambient conditions, reducing environmental impact [6,7]. Plant-extract-mediated green synthesis has gained attention due to its eco-friendly nature that utilizes various parts of plants as a natural reducing and capping agent [8,9]. Such decorated NPs not only minimize toxicity but also support the advancement of green chemistry. Moreover, green synthesis is cost-effective, resource-efficient, and time-saving compared to conventional methods [10–15]. Capping agents, including citrate molecules (Cit), 1,5-diphenyl-1,3,5-pentanetrione (Pent), and dimethyl-L-tartrate (DMLT), retain the structural integrity of pristine and doped metal oxides and peroxides for more than one month [16–18]. Similarly, several phytochemicals in plant extracts act as capping agents to control NP growth. The choice of plants depends on various factors, including availability, history, economical, and bio-compatibility. The use of medicinal plants, plants which have been practiced for centuries for various kinds of treatments, is attributed to enhancing the therapeutic potential of plant extract decorated NPs [19].

The increasing microbial resistance to antibiotics has driven research toward investigating alternative sources for treating resistant strains [20,21]. Approximately 80% of the world's population relies on plant-derived medicines as a first line of defense in maintaining health and combating disease. Thousands of plants currently serve as remedies for various ailments. The biomolecules in the extract, including antioxidants, effectively reduce metal ions to their NPs counterparts and stabilize

them by preventing aggregation [22–24]. Lots of medicinal antiviral enzymes inhibitory and antibacterial as been studied for their biological activities [25,26].

Medicinal plants have great medicinal value and the *Ficus benghalensis* (Banyan tree) is one of them. It is a well-known medicinal plant in Khyber Pakhtunkhwa (KPK), Pakistan, and is locally utilized for alleviating conditions such as pain, fever, inflammation, insomnia, diabetes, wound healing, and for its antioxidant and antimicrobial properties [27]. Singh, Dhankhar et al (2023) investigated the antimicrobial activity and phytochemical composition of Ficus benghalensis leaf and fruit extracts, demonstrating the presence of bioactive compounds such as lupeol, beta amyrone, and vitamin E, which contribute to its therapeutic potential against bacterial and fungal infections [28]. In another study, Torane et al. (2020) evaluated the antimicrobial and antifungal potential of Ficus benghalensis aerial part extracts prepared under different temperature conditions, confirming significant activity against bacterial strains such as *Staphylococcus aureus* and *Escherichia coli*, as well as fungal strains like *Candida albicans* and *Aspergillus niger* [29]. Similarly, Murugesu et al. (2021) comprehensively reviewed the phytochemical profile and pharmacological properties of *Ficus benghalensis* and *Ficus religiosa*, highlighting their diverse bioactive compounds such as flavonoids, alkaloids, and terpenoids. The study also emphasized their broad-spectrum biological activities, including antioxidant, anticancer, antimicrobial, and wound healing properties, along with recent applications in nanotechnology [27]. Due to this diverse array of phytochemicals and biological activities, the *Ficus bengalinsis* is a suitable codidate of the synthesis of Fe NPs. Moreover, Its rapid growth, renewable biomass, and bioactive phytochemicals contribute to an eco-friendly approach to nanoparticle synthesis, reducing the reliance on hazardous chemicals. Utilizing plant-based synthesis not only minimizes environmental toxicity but also aligns with sustainable development goals by promoting biodegradable and non-toxic alternatives for biomedical and environmental applications. Silver, a noble metal with well-known antimicrobial and antifungal properties, has been utilized since ancient times. The potential of Ag NPs increases significantly at the nanoscale and has been explored over the past decade in various fields, including medical imaging, targeted drug delivery, and enzyme inhibition. The most important effect is the surface plasmon resonance (SPR) effect, which makes Ag NPs particularly valuable in biomedical applications [30–33]. Enzymes, which act as biological catalysts, facilitate various biochemical reactions. Their inhibitors can treat several neurological disorders, including those without proper treatment, such as acetylcholinesterase, tyrosinase, xanthine oxidase, alkaline phosphatase, and glucosidase. However, enzymes can be inhibited by pesticides or toxic substances, which disrupt normal physiological functions and cause severe side effects. Therefore, reliable methods to detect enzymatic inhibitors are essential, considering their potential correlation with medical conditions like gout, melanin hyperpigmentation, and Alzheimer's disease [34–38]. The effectiveness of pharmaceutical compounds is linked to their impact on enzymatic activity [39]. Specific toxins, including extracts and NPs, can irreversibly inhibit enzymes, affecting biological processes and health. Metallic NPs can interact with enzymes, disrupting functional groups or active sites, and reducing catalytic function. Understanding these molecular interactions is crucial for precision diagnostics and understanding NPs interactions in various scientific domains [40–42]. Silver crystallizes in a face-centered cubic (FCC) with space group Fm-3m, where each Ag atom is bounded to 12 nearest neighbors in an octahedral arrangement. The arrangement forms a mixture of edge, face and corner-sharing octahedra. High electrical and thermal conductivity, malleability, and reflectivity are inherited due to silver's metallic bonding and dense atomic packing. These properties make it ideal in electronics, jewelry, and catalysts. Additionally, its FCC structure provides low thermal expansion [43].

Despite the promising potential of green-synthesized silver nanoparticles, certain limitations and challenges remain. The variability in plant extract composition due to environmental factors can lead to inconsistencies in nanoparticle synthesis and bioactivity. Additionally, the precise mechanisms governing the interaction of biomolecules with silver ions during synthesis are not yet fully elucidated. Challenges also exist in scaling up the green synthesis process for industrial applications while maintaining reproducibility and stability. Further studies are needed to optimize synthesis conditions, ensure long-term stability, and assess potential cytotoxicity for safe biomedical applications. Addressing these challenges will help advance the practical applications of green-synthesized AgNPs in diverse fields.

Although extensive research has been conducted on the biomedical applications of metallic nanoparticles, there remains a critical gap in understanding the molecular mechanisms underlying their enzyme-inhibitory activity. Most studies focus on the synthesis and basic biological evaluations of Ag NPs, but the detailed molecular interactions between these nanoparticles and target enzymes remain unexplored. The lack of molecular docking studies to elucidate the binding affinity, and active site interactions, induced by Ag NPs limits our understanding of their mechanistic action. Additionally, the role of functional biomolecules from Ficus benghalensis in stabilizing Ag NPs and enhancing their bioactivity at the molecular level is not well established. This study addresses these gaps by utilizing molecular docking analysis to predict the interaction of Ag NPs with key metabolic enzymes, such as urease, α-glucosidase, carbonic anhydrase II, and xanthine oxidase. By investigating binding energies and hydrogen bonding interactions, this study provides mechanistic insights into the therapeutic potential of green-synthesized Ag NPs. Furthermore, *In Vivo* pharmacological assessments complement the computational findings, offering a comprehensive evaluation of their biomedical relevance. The findings highlight the promising biomedical applications of *Ficus benghalensis* root extracts (FBRE) and Ag NPs as novel enzyme inhibitors. These findings opened the way for targeted therapeutic interventions and demonstrated significant analgesic and sedative effects, underscoring the potential of these compounds for future drug development. Furthermore, this suggests that FBR extract and Ag NPs could be valuable in developing new treatments for various conditions.

## 2. Materials and methods

### 2.1. Ethics

The animal study results were obtained according to guidelines, regulations, and institutional policies. The animals were stored under standard laboratory conditions. The animal study was approved by the ethical committee of the Department of Pharmacy, Abdul Wali Khan University Mardan, with Ref. No of EC/DOP/12. All the selected animals were properly acclimatized with laboratory conditions. After suitable acclimatization, each animal was treated with the approved route of administration. Once the experiment was completed, animals were killed with cervical dislocation, following approved ethical guidelines.

### 2.2. Plant collection and extraction

*Ficus benghalensis* was collected from Peshawar, Pakistan, and identified by Mr. Saifullah, a taxonomist at Government Degree College Lahore Swabi. The root material was dried at room temperature for 7 days, ground, and extracted using a water-methanol solvent system for 4 days. After filtration, the extract was concentrated via rotary evaporation and further dried using a water bath at 50°C to yield a solvent-free, bioactive-rich extract for subsequent use.

### 2.3. Synthesis of nanoparticles

Ag NPs were synthesized via a green method using *Ficus Benghalensis* root extract (FBRE). A 1 $mM$ silver nitrate ($AgNO_3$) the solution was prepared by 42.46 $mg$ of silver $AgNO_3$ in 250 $mL$ of di-ionized water (DIW). FBRE was prepared by dissolving 0.1 $g$ of the methanolic extract powder in 100 $mL$ of methanol and diluting it to 500 $mL$ with DIW. The extract was then filtered. The reaction mixture was optimized by testing different salt-to-extract ratios (1:1, 1:2, 1:4, 1:6, 1:8, 1:10) under continuous stirring. A color change from the initial solution to yellowish-brown, observed at a 1:8 ratio, served as a preliminary indication of Ag NPs formation. To separate the NPs, the reaction mixture was centrifuged at 3000 rpm for 40 minutes, followed by washing with double-distilled water and vacuum drying. Confirmation of Ag NPs synthesis was achieved through UV-Vis spectroscopy.

### 2.4. Instrumentations

The synthesized Ag NPs were thoroughly characterized using various techniques at the Material Research Laboratory (MRL), Department of Physics, University of Peshawar. Scanning Electron Microscopy (SEM) with National Center of

Excellence in Geology, University of Peshwar (NCEG-UOP) facilitated the visualization of the NPs' surface morphology and topography. Functional groups potentially attached to the NPss were identified using Fourier Transform Infrared Spectroscopy (FTIR). UV-Vis spectroscopy was employed to determine the absorption, while Photoluminescence (PL) spectroscopy investigated the luminescent properties of the Ag NPs. Notably, the observed Surface Plasmon Resonance (SPR) effect in the UV-Vis spectrum further supported the spherical morphology of the synthesized Ag NPs.

## 2.5. Computational studies

Using the Spanish Initiative for Electronic Simulations with Thousands of Atoms (SIESTA), density functional theory (DFT) calculations were carried out for the Ag (2x2x1) layer's structural, thermodynamic, mechanical, optoelectronic, and optical properties. We used the Perdew-Burke-Ernzerhof (PBE) in conjunction with the generalized gradient approximation (GGA) to account for the exchange-correlation effects. The interaction between the valence electron and the core was described using Troullier-Martins pseudopotentials that conserve norms. From the material project database, an Ag crystallographic information file (cif) was acquired. The unit cell was extended along the x- and y-axes (2x2x1) to form a supercell with a 10 Å vacuum along c-axis to avoid any contact between periodic images in that direction. In the supercell, 24 Ag atoms were found. Fig 1 illustrates the relaxed geometry of such a system with bond length before and after optimization, yielding values of 2.9 Å and 2.96 Å. Before calculations, a convergence test was performed using the mesh cut-off (Ry), kpoint grid, and lattice constant (Å). The overall energy convergence criteria were set at $10^{-6}$ eV, and geometry modifications were made until the stresses on each atom were less than 0.01 eV/Å. The atomic orbits were described using the double-zeta polarized (DZP) basis set. To guarantee that the charge density and potentials were integrated with the appropriate degree of accuracy, a mesh cut-off energy of 350 Ry was applied to the real-space grid. To ascertain mechanical stability (various models are fitted to the volume vs. total energy data) and optoelectronic properties (density of states, projected density of states, band structures, and optical properties were examined), cohesive energy (eV/atom) and enthalpy of formation (eV/atom) were calculated for an Ag system. Similarly, the molecular docking was also performed to investigate the binding interaction of Ag NPs with the targeted enzymes, including carbonic anhydrase II (CA-II), urease, xanthine oxidase, and α-glucosidase, which were assessed via AutoDock Vina. The selected enzymes were downloaded from the protein data bank. At the first step, water molecules and ligands were removed, followed by adding polar hydrogen and Kollman charges. The molecular docking studies signify a strong interaction of Ag NPs with the targeted enzymes.

## 2.6. *In vitro* biological screening

### 2.6.1. Urease inhibition assay.
The urease inhibition assay was performed using the modified Berthelot method, which quantifies ammonia production from the hydrolysis of urea. In a 96-well microplate, 10 μL of the synthesized Ag NPs (0.2 μg) or *Ficus benghalensis* root extract (FBRE, 0.2 μg) was mixed with 25 μL of Jack bean urease enzyme

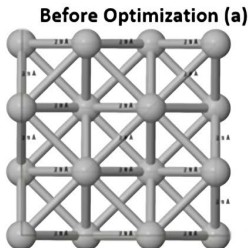
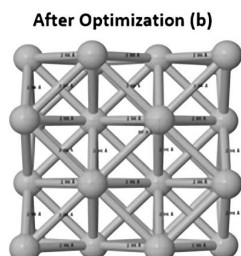

**Fig 1. Layer structure of silver before (a) and after optimization (b).**

solution (1 U/mL) and 40 µL of phosphate buffer (pH 7.4). After a 15 min pre-incubation at 37°C, 40 µL of urea (100 mM) was added as the substrate, and the reaction was incubated for 30 minutes. Thiourea (0.2 µM) was used as the standard urease inhibitor. The ammonia produced was quantified by adding 40 µL each of phenol reagent (1% phenol, 0.005% sodium nitroprusside) and hypochlorite reagent. Absorbance was measured at 630 nm, and the inhibition percentage was calculated relative to the control [44].

**2.6.2. α-Glucosidase inhibition assay.**  The α-glucosidase inhibition activity was assessed to determine the efficacy of the synthesized Ag NPs (0.2 µg) and FBRE (0.2 µg) in inhibiting enzyme activity. Each sample (10 µL) was pre-incubated with 25 µL of α-glucosidase enzyme solution (0.1 U/mL) in a 96-well plate for 15 minutes at 37°C. p-Nitrophenyl-α-D-glucopyranoside (PNPG, 1 mM) was added as the substrate, and the reaction was allowed to proceed for 30 min. The reaction was stopped by adding 40 µL of sodium carbonate solution (100 mM), and absorbance was measured at 405 nm to determine the release of p-nitrophenol. Acarbose (0.2 µM) was used as the standard α-glucosidase inhibitor, with inhibition percentage calculated against the control [45].

**2.6.3. Carbonic Anhydrase II (CA-II) inhibition assessment.**  The CA-II inhibition assay was conducted using a colorimetric method to measure the inhibition of CA-II by the synthesized Ag NPs (0.2 µg) and FBRE (0.2 µg). A reaction mixture containing 10 µL of sample and 25 µL of CA-II enzyme solution (0.2 U/mL) was pre-incubated in a 96-well plate for 15 minutes at 37°C. The reaction was initiated by adding 40 µL of p-nitrophenyl acetate (1 mM) as the substrate and allowed to proceed for 30 min. Absorbance was recorded at 405 nm to monitor the production of p-nitrophenol. Acetazolamide (0.2 µM) was used as the standard CA-II inhibitor, with inhibition percentage calculated relative to the control [46].

**2.6.4. Xanthine oxidase (XO) inhibition assay.**  The xanthine oxidase inhibition assay evaluated the ability of the synthesized Ag NPs (0.2 µg) and FBRE (0.2 µg) to inhibit xanthine oxidase. In a 96-well plate, 10 µL of each sample was combined with 40 µL of xanthine oxidase enzyme solution (0.1 U/mL) and 50 µL of phosphate buffer (pH 7.5). After a 10 min pre-incubation at 25°C, 50 µL of xanthine (100 µM) was added as the substrate. The reaction was incubated for 30 minutes, and absorbance was measured at 290 nm to quantify uric acid production. Allopurinol (0.2 µM) was used as the standard XO inhibitor, and inhibition percentage was calculated in relation to the control [44].

## 2.7. *In vivo* activities

**2.7.1. Analgesic activity.**  The analgesic potential was evaluated by acetic acid-induced writhing model. In this procedure animals were classified as the negative control group which was administered with distilled water (10 ml/kg), the positive control group was treated with diclofenac (10 mg/kg) and the tested groups were administered with extract (15, 25, 50 and 100 mg/kg) and Ag NPs (2.5, 5 and 10 mg/kg). After 30 min post-treatment, each animal was injected with 1% acetic acid solution (IP). After 10 min of acetic acid treatment, each animal was observed individually for abdominal constrictions (writhes) for 5 min. the percent analgesic effect was calculated using the following formula [44].

$$analgesic\ activity\ = 100 - \frac{Number\ of\ writhings\ in\ tested\ animals}{Number\ of\ writings\ in\ control\ animals} \times 100$$

**2.7.2. Sedative activity.**  The samples were tested for sedative effect in an open field *in-vivo* model. This experiment was conducted following our published procedure with modifications. The animals were classified as above, and the positive control group was injected with diazepam (0.5 mg/kg). The same doses for both samples were used. A special wooden box was used. The bottom board of the box was lined with equal spaces. After 30 min of the treatment with distilled water, diazepam, and extract/ NPs the animal was placed at the center of the box and the number of lines crossed was counted for 10 min [47].

## 3. Results

### 3.1. Structural, thermal and mechanical properties

The thermodynamic and mechanical stability of the simulated Ag 2x2x1 layer with 10 Å vacuum in the c-direction were measured using cohesive energy (eV/atom), enthalpy formation (eV/atom), and Murnaghan, Birch-Murnaghan, Birch, and Vinet models. Cohesive energy is defined, as the energy required to disassemble the material into individual atoms, and enthalpy formation reflects the change energy to form a compound from its constituent elements. The calculation shows, cohesive energy of −2.520 eV/atom and enthalpy formation of −0.678 eV/atom, summarized in Table 1. The results indicate a stable configuration and suggest that the system has strong interatomic bonds. To look into the response of Ag Nps to compressive deformation, we fitted various kinds of models to the relative energy vs. Volume data, shown in Fig 2. The Murnaghan, Birch-Murnaghan, Birch, and Vinet models resulted in equilibrium energy ($E_0$ (eV)), a bulk modulus, a pressure derivative of the bulk modulus, and equilibrium volume with value of −23621.612 eV, −23621.911 eV, −23621.911 eV and −23622.002 eV, 54.39 Gpa, 63.03 Gpa, 63.03 Gpa, and 65.54 Gpa, 5.36, 5.63, 5.63, and 5.67, 537.61 Å³, 533.42 Å³, 533.42 Å³, and 532.74 Å³ respectively as shown in the inset of Table 1. The relatively high bulk modulus across all models confirms the mechanical robustness of the simulated Ag layers. The small differences between the fitted models also highlight that the NPs structure remains consistent and stable under different theoretical frameworks, with the Vinet model suggesting the least compressibility.

### 3.2. Optoelectronic properties

The total density of states (TDOS), the projected density of states (PDOS), the band plot, and optical properties were calculated for Ag (2x2x1) supercell. The total density of states confirmed the metallic behavior of Ag with a significant peak

**Table 1. Cohesive energy and enthalpy formation of silver layer.**

| Material | Total Energy (eV) | Energy of Silver atom | Energy of Silver atom in bulk | Cohesive Energy (eV/atom) | Ethalpy Formation (eV/atom) |
|---|---|---|---|---|---|
| Silver (Ag) 2x2x1 layer | −23621.913 | −981.726 | −983.568 | −2.520 | −0.678 |

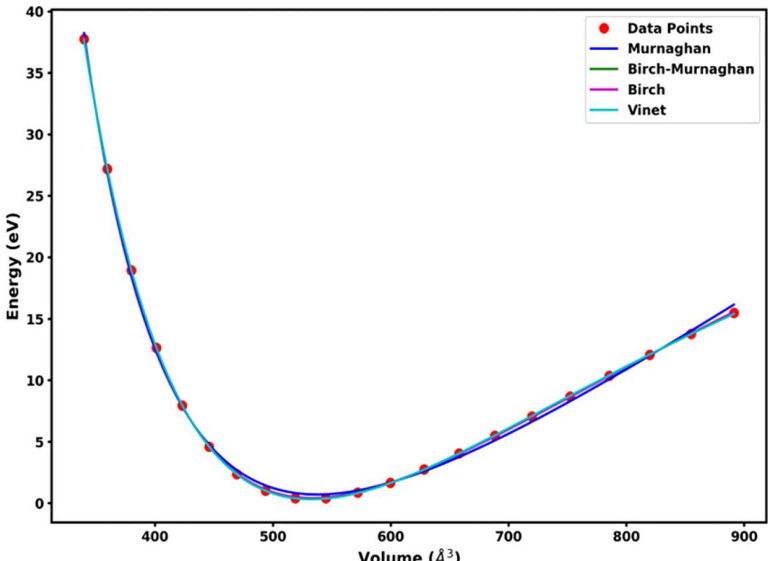

**Fig 2. Various Equation of state fit to the volume vs. relative energy of Ag 221 layer.**

around −4 eV, attributed to the Ag-4d orbital, which is the main contributor to the DOS near the Fermi level ($E_f$). The Ag-5s states have a contribution to the conduction band. The analysis provides insights into the electronic structure of Ag. Such results play an important role in understanding the conductive properties. The same behavior is shown in the band plot. The optical properties show sufficient absorption in the visible spectrum, as shown in the panel of absorption coefficient vs. energy. In the conductivity vs. energy plot, a significant peak at lower energies, signifying a high conductive response near the Fermi level. These optical characteristics are consistent with the known behavior of silver, and such analyses provide valuable insights into the electronic structure and the optoelectronic properties of metallic thin films. The results are shown in Fig 3a–c.

### 3.3. UV-Vis spectroscopy and surface plasmon resonance (SPR) of Ag NPs

Silver NPs exhibit a fascinating property known as the Surface Plasmon Resonance (SPR) effect. This phenomenon occurs when light interacts with the free electrons on the Ag NPs surface. The specific wavelength of light excites these electrons, causing a collective oscillation and leading to the absorption and subsequent re-emission of light. Notably, the absorption peak is highly dependent on the size and shape of the NPs. In this study, the UV-Vis spectrum (Fig 4) revealed a broad absorption peak at around 430 nm, which is a characteristic signature of the SPR effect in spherical Ag NPs. This observation not only confirms the presence of Ag NPs but also strongly supports their spherical morphology. The observed behavior aligns with the established relationship between the SPR peak and NPs shape.

### 3.4. FTIR spectroscopy

FTIR analysis (Fig 5) revealed the functional groups present in *Ficus Benghalensis* root extract (FBRE) which reduced and caped the synthesized Ag NPs, recorded in the range of 500−4000 cm⁻¹. The FBRE spectrum exhibited peaks indicative of $C-O$ stretching in carbohydrates (887 cm⁻¹, 1027 cm⁻¹, 1067 cm⁻¹) and aromatic compounds (1222 cm⁻¹, 1621 cm⁻¹),

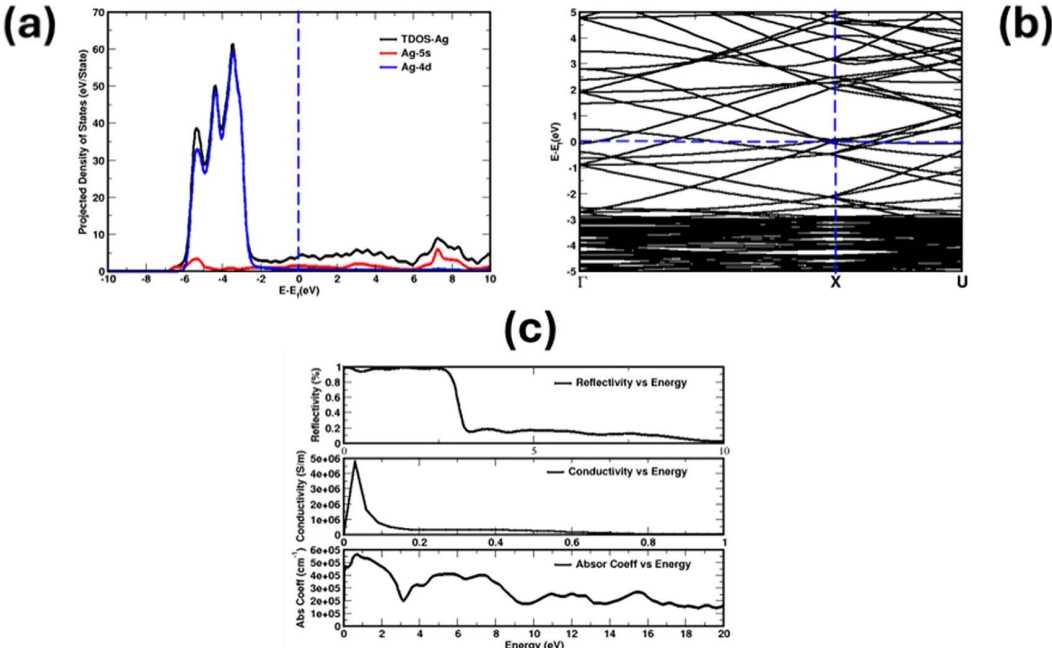

**Fig 3. Total density of States, Projected density of States (a), Band (b) and optical properties (c) of Ag 221 layer.**

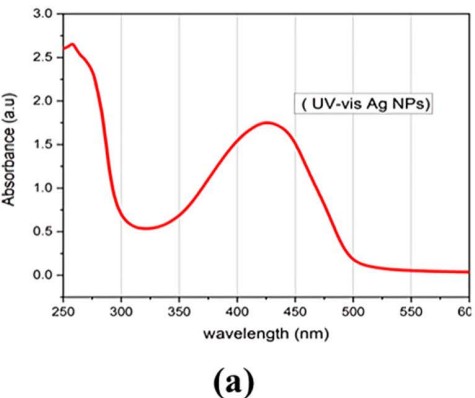
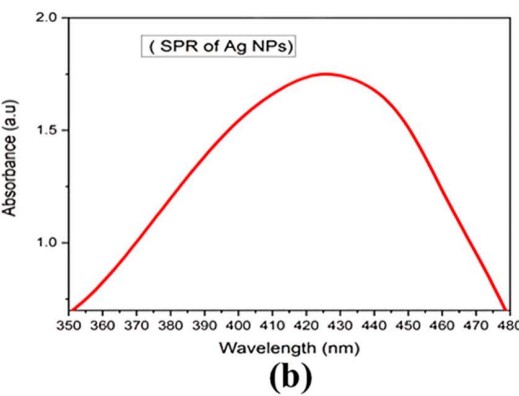

**Fig 4. UV-visible spectra (a), and surface plasmon resonance of Ag NPs (b).**

alongside $C-H$ stretching in alkanes and lipids (2940 cm⁻¹, 2981 cm⁻¹), and $O-H$ stretching in alcohols/water (3362 cm⁻¹). The Ag NPs spectrum displayed similar $C-O$ vibrations (1045 cm⁻¹, 1115 cm⁻¹), but also showcased $C-N$ stretching (1268 cm⁻¹), $C-H$ bending (1394 cm⁻¹, 1588 cm⁻¹), carbon-carbon triple bond stretching (2167 cm⁻¹), and $C-H$ stretching in proteins (2851 cm⁻¹, 2927 cm⁻¹), and $O-H$ stretching (3248 cm⁻¹). The presence of overlapping functional groups in both spectra suggests their involvement in Ag NPs synthesis and stabilization. However, peak shifts in the Ag NPs spectrum compared to FBRE indicate an interaction between the plant extract and silver ions. Notably, peaks at 1394 cm⁻¹, 1588 cm⁻¹, and 1621 cm⁻¹ present in both samples suggest aromatic compounds, potentially playing a role in Ag NPs reduction and capping. These findings support the role of FBRE biomolecules in Ag NPs synthesis and stabilization.

### 3.5. Scanning electron microscopy (SEM)

SEM analysis (JSM-IT100, National Centre of Excellence in Geology, University of Peshawar (NCEG, UoP)) revealed a slight size distribution of spherical Ag NPs synthesized using *Ficus Benghalensis* root extract. The SEM images of single and cluster of Ag NPs as shown in Fig 6a,b. The average diameter particle 41.55 nm size, determined using ImageJ and plotted in Origin Lab Fig 6c, was, with most particles falling within the 20–55 nm range. Particles exhibit a uniform spherical shape, both single and clusters are observed, indicating controlled synthesis. The cluster suggests partical-partical interactions. Surface morphology influences optical and biological properties The observed size distribution and spherical shape suggest the plant extract acted as a capping agent, potentially interacting with the Ag NPs to control aggregation.

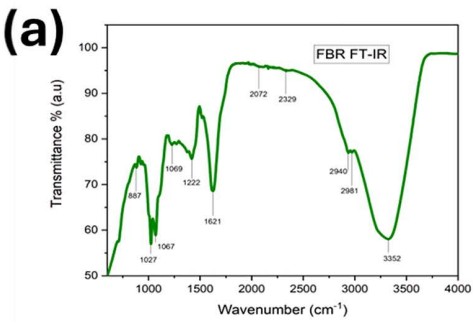
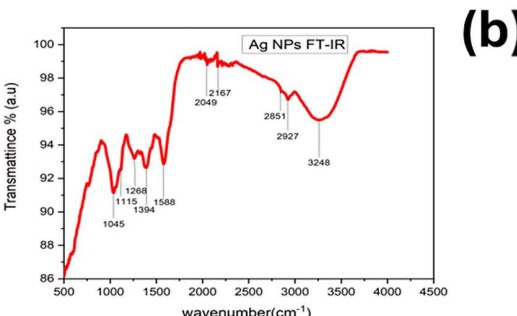

**Fig 5. FT-IR spectra of FBRE (a) and Ag NPs (b).**

These findings from SEM analysis confirm the successful green synthesis of Ag NPs with promising characteristics for further exploration.

### 3.6. Energy dispersive X-ray spectroscopy

Energy-dispersive X-ray Spectroscopy (EDX) analysis (Fig 7) confirmed the elemental composition of the green-synthesized Ag NPs derived from *Ficus Benghalensis* Root (FBR) extract. The EDX spectrum revealed distinct peaks for silver (Ag), oxygen (O), potassium (K), and surprisingly, chlorine (Cl). While the presence of Ag confirms the purity of NPs, the other elements suggest the involvement of organic compounds. These likely originate from biomolecules in the FBR extract, particularly polyphenols and other carbon-containing molecules. Notably, the EDX analysis provides valuable information about the elemental composition near the surface of the Ag NPs, offering insights into potential interactions between the NPs and the biomolecules during synthesis.

### 3.7. Pharmacological application

**3.7.1. Urease inhibition.** The effect of FBRE and Ag NPs on urease is demonstrated in Table 2. FBRE (0.2 µg) and Ag NPs (0.2 µg) showed 57.98% and 80.76% inhibitory effects, respectively, with $IC_{50}$ values of $44.87 \pm 1.11$ µM and $35.76 \pm 1.20$ µM. Thiourea used as a standard drug, exhibited a 98.93% inhibitory effect with an $IC_{50}$ value of $21.38 \pm 0.82$ µM.

**3.7.2. Effect on α-glucosidase activity.** Both of the tested samples showed a variable degree of α-glucosidase inhibitory effect (Table 2); however, the effect of Ag NPs was more significant than FBRE. The percent inhibitory action of

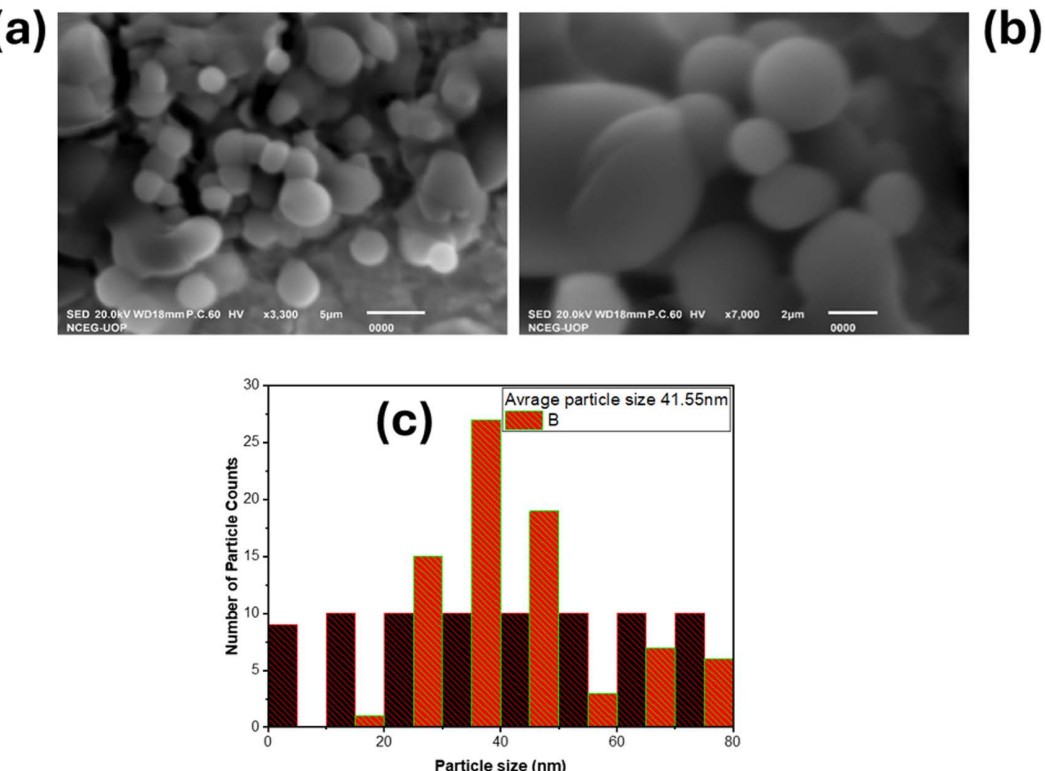

**Fig 6. SEM images of single and cluster of Ag NPs with low (a), high (b) resolutions and size distribution curve (c).**

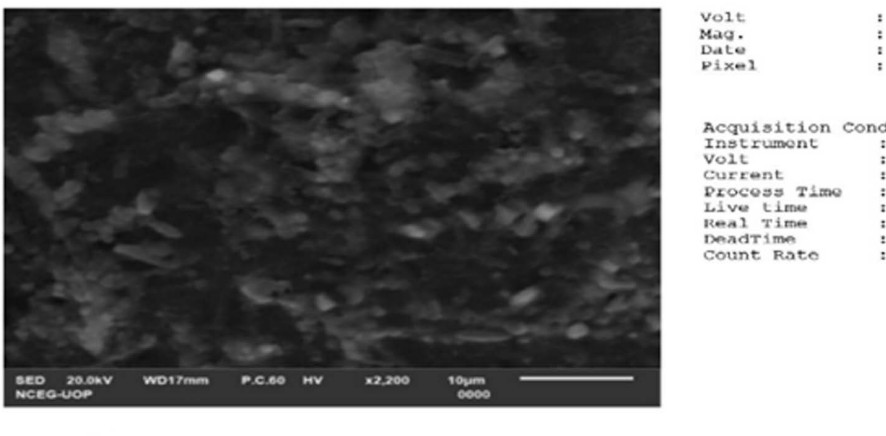

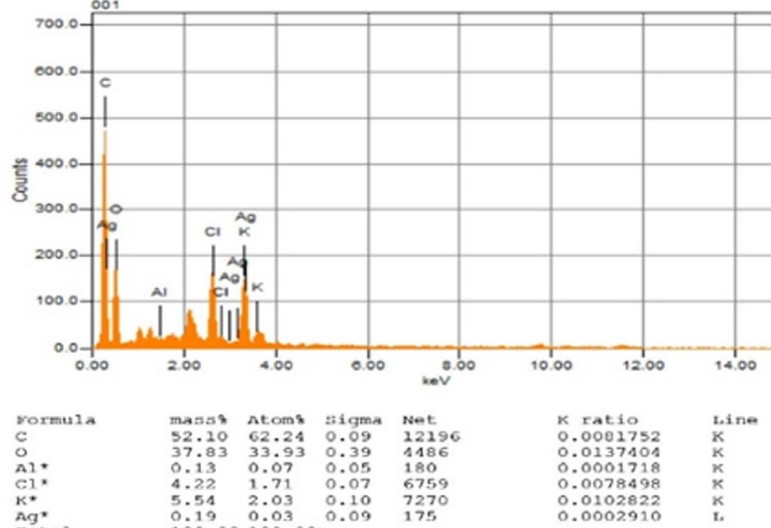

**Fig 7. EDX spectrum of synthesized Ag NPs.**

FBRE (0.2 µg) and Ag NPs (0.2 µg) was 37.09% and 80.98%, respectively. The $IC_{50}$ value of Ag NPs was $90.65 \pm 1.54$ µM, while the standard inhibitor, acarbose, showed 98.54% inhibition with an $IC_{50}$ value of $28.98 \pm 1.00$ µM.

**3.7.3. Effect on carbonic anhydrase II (CA-II).** The extract and Ag NPs significantly inhibited CA-II activity. FBRE (0.2 µg) exhibited a 70.43% inhibitory effect with an $IC_{50}$ value of $0.24 \pm 0.54$ µM, whereas Ag NPs (0.2 µg) showed a higher inhibition of 89.32% with an $IC_{50}$ value of $0.19 \pm 0.03$ µM. Acetazolamide, the standard inhibitor, showed 92.33% inhibition with an $IC_{50}$ value of $0.15 \pm 0.04$ µM (Table 2).

**3.7.4. Effect on xanthine oxidase (XO).** The effect of FBRE and Ag NPs on XO activity is shown in Table 2. FBRE (0.2 µg) exhibited a 30.55% inhibitory effect, while Ag NPs (0.2 µg) showed a 49.90% inhibitory effect. Allopurinol, the standard inhibitor, demonstrated 98.54% inhibition with an $IC_{50}$ value of $2.65 \pm 0.04$ µM.

**Table 2. Enzyme inhibitory potential of FBRE and Ag NPs.**

| Enzyme | Sample | Conc. | % inhibition | $IC_{50}$ |
|---|---|---|---|---|
| **Urease** | FBE | 0.2 µg | 57.98 | 44.87 ± 1.11 |
| | Ag NPs | 0.2 µg | 80.76 | 35.76 ± 1.20 |
| | Thiourea | 0.2 µM | 98.93 | 21.38 ± 0.82 |
| **α-glucosidase** | FBE | 0.2 µg | 37.09 | – |
| | Ag NPs | 0.2 µg | 80.98 | 90.65 ± 1.54 |
| | Acarbose | 0.2 µM | 98.54 | 28.98 ± 1.00 |
| **Carbonic anhydrase II** | FBE | 0.2 µg | 70.43 | 0.24 ± 0.54 |
| | Ag NPs | 0.2 µg | 89.32 | 0.19 ± 0.03 |
| | Acetazolamide | 0.2 µM | 92.33 | 0.15 ± 0.04 |
| **Xanthine Oxidase** | FBE | 0.2 µg | 30.55 | – |
| | Ag NPs | 0.2 µg | 49.90 | – |
| | Allopurinol | 0.2 µM | 98.54 | 2.65 ± 0.04 |

**3.7.5. Analgesic activity.** The analgesic potential was evaluated using the acetic acid-induced writhing model as shown in Table 3. FBRE exhibited dose-dependent inhibition, with values of 32.76% (15 mg/kg), 40.98% (25 mg/kg), 61.19% (50 mg/kg), and 68.09% (100 mg/kg). Ag NPs also showed significant dose-dependent inhibition, achieving 72.98% (2.5 mg/kg), 78.23% (5 mg/kg), and 83.09% (10 mg/kg). Diclofenac sodium (10 mg/kg), used as a positive control, demonstrated 84.01% inhibition.

## 3.8. Sedative activity

FBRE showed dose-dependent activity with reductions in movement of 60.16% (15 mg/kg), 52.09% (25 mg/kg), 43.32% (50 mg/kg), and 36.66% (100 mg/kg). Ag NPs also showed a dose-dependent reduction in movement, with values of 34.65% (2.5 mg/kg), 25.44% (5 mg/kg), and 16.09% (10 mg/kg). Diazepam (0.5 mg/kg), the positive control, exhibited 8.60% inhibition (Table 4).

## 3.9. Molecular docking

Docking analysis suggests robust and long-lasting interactions, including hydrogen bonding, hydrophobic interaction, electrostatic interaction and π-π stacking. The functional groups that capped and stabilized Ag NPs also facilitated interactions

**Table 3. Analgesic potential of extract and its synthesized NPs from *Ficus benghalensis*.**

| Treatment | Dose (i.p) | % Inhibition of writhing |
|---|---|---|
| **Saline** | 10 mL/kg | – |
| **Diclofenac Sodium** | 10 mg/kg | 84.01 ± 0.82*** |
| **FBE** | 15 mg/kg | 32.76 ± 2.01 |
| | 25 mg/kg | 40.98 ± 1.87* |
| | 50 mg/kg | 61.19 ± 1.90* |
| | 100 mg/kg | 68.09 ± 1.87** |
| **Ag NPs** | 2.5 mg/kg | 72.98 ± 1.23** |
| | 5 mg/kg | 78.23 ± 1.40*** |
| | 10 mg/kg | 83.09 ± 1.65*** |

*p < 0.05; **p < 0.01; ***p < 0.001

**Table 4. Sedative potential of extract and its synthesized NPs from *Ficus benghalensis*.**

| Treatment | Dose (i.p) | % Inhibition of writhing |
|---|---|---|
| **Saline** | 10 mL/kg | – |
| **Diazepam** | 0.5 mg/kg | 8.60±0.88*** |
| **FBE** | 15 mg/kg | 60.16±2.34 |
| | 25 mg/kg | 52.09±1.09* |
| | 50 mg/kg | 43.32±1.66* |
| | 100 mg/kg | 36.66±1.90** |
| **Ag NPs** | 2.5 mg/kg | 34.65±1.55** |
| | 5 mg/kg | 25.44±1.34*** |
| | 10 mg/kg | 16.091.09*** |

$*p<0.05$; $**p<0.01$; $***p<0.001$

with the target enzymes, including carbonic anhydrase II, urease, xanthine oxidase, and α-glucosidase, through various bonds. Based on the FTIR spectra, the functional groups present on the surface of NPs can interact with the enzymes, contributing to the observed activities. Functional groups such as C-O and C-H interacted with residues like ARG27, PHE131, GLY132, and LYS133 in carbonic anhydrase II through hydrogen bonding and van der Waals interactions, stabilizing the NPs-enzyme complex and potentially disrupting enzymatic activity (Fig 8a). In the case of urease, O-H and C-N functional groups formed hydrogen bonds with active site residues such as HIS409, TYR410, and GLN414, while aromatic compounds interacted with residues like LEU415 and ARG439 via π-π stacking, suggesting competitive or non-competitive inhibition of the enzyme (Fig 8b). Similarly, the aromatic and hydroxyl groups formed hydrogen bonds and hydrophobic interactions with residues such as ARG381, THR396, HIS614, and ALA615 in xanthine oxidase, likely interfering with the enzyme's active site and reducing its catalytic efficiency (Fig 8c). For α-glucosidase, the hydroxyl and carbonyl groups formed hydrogen bonds with residues like TYR14, ILE16, and ASP23, while aromatic compounds interacted with PHE21 and LEU22, demonstrating strong binding affinity and contributing to enzyme inhibition (Fig 8d). These findings suggest that the functional groups from FBRE not only mediated the synthesis and stabilization of Ag NPs but also enhanced their interactions with enzymes, leading to potential inhibitory effects on enzymatic activity. This dual role of biomolecules from FBRE highlights their significance in both nanoparticle synthesis and biological applications.

## 4. Discussion

The Ag NPs were synthesized using a green synthesis approach. The extract served as a capping and stabilizing agent, ensuring the chemical and physical stability of the nanoparticles and preventing their degradation. Currently, there are several chemical and physical techniques used for the synthesis of Ag NPs [11]. However, green synthesis of Ag NPs is a very simple and eco-friendly synthetic route for the synthesis of Ag NPs. These green synthesis techniques aim to address the biocompatible good capping agent providing extract of plant and sustainability concerns associated with conventional synthesis approaches [48,49]. In this regard, *Ficus Benghalensis* root extract FBRE proves a very important biological substrate in the right direction for the green synthesis of Ag NPs [50]. The SPR study and SEM images confirm that Ag NPs spherical in shape and equally distributed. The SPR absorption band it 430 nm strongly observed. The SEM images show that the synthesized Ag NPs are spherical. The particles are not dispersed but seem agglomerated due to the presence of some biomolecules that provide a capping and stabilizing agent to Ag NPs. The biomolecules present in the extract of plants are responsible extraction of Ag NPs. In the present research study, the particle size of Ag NPs was found in different ranges but the average range of 41.55nm. The surface morphology of the particle formed consists of Ag NPs spherical with (FCC) with space group Fm-3m. The XRD pattern is clear that the synthesized NPs by the green

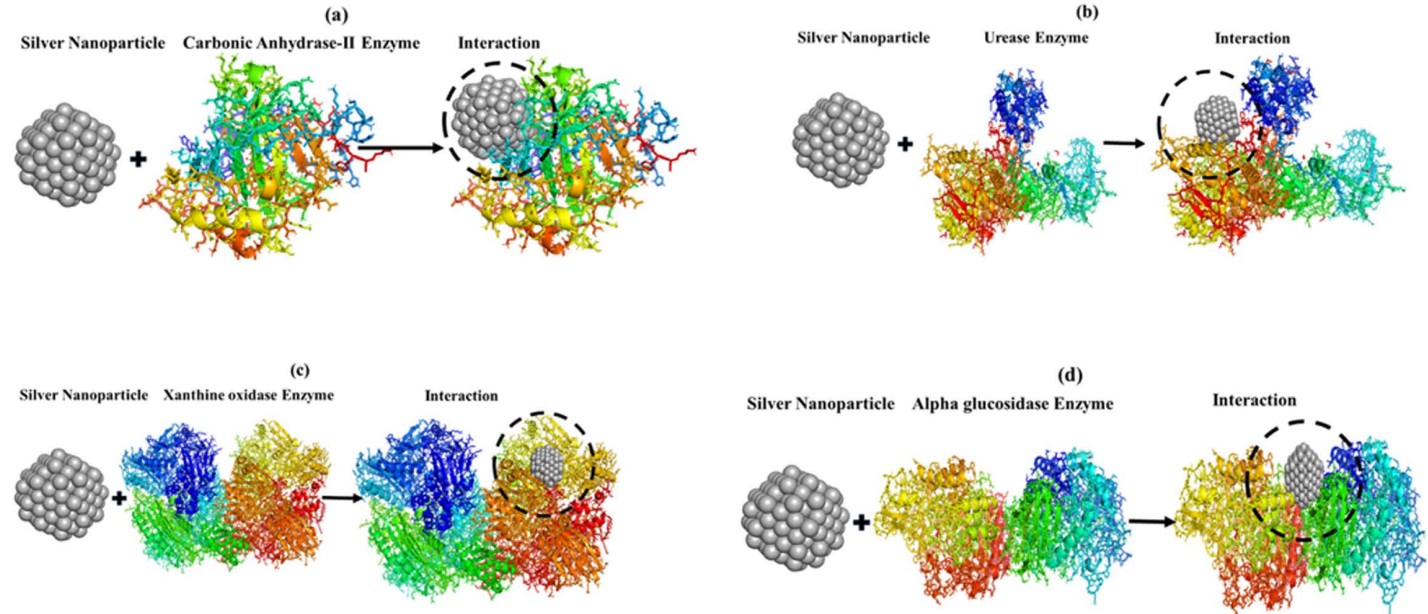

**Fig 8. Molecular docking of Carbonic anhydrase-II Enzyme (a), Urease Enzyme (b), Xanthine oxidase Enzyme, and α-glucosidase Enzyme.**

synthetic method are highly crystalline in nature and stable. This study will there for conform to the development and easy bioprocess for the synthesis of Ag NPs from FBRE and open a new possibility of synthesizing Ag NPs from natural products which will be useful in biomedical applications. The biological activities of the synthesized Ag NPs were evaluated in vitro through enzyme inhibition assays, where they exhibited significant inhibitory effects on urease, α-glucosidase, carbonic anhydrase II (CA-II), and xanthine oxidase (XO). Among these enzymes, Ag NPs showed the highest inhibition against CA-II (89.32%) and α-glucosidase (80.98%), surpassing the inhibitory effects of FBRE and demonstrating their potential as enzyme inhibitors. The Ag NPs exhibited an $IC_{50}$ value of 0.19 µM against CA-II, which is comparable to acetazolamide, a standard CA-II inhibitor. Similarly, Ag NPs exhibited an $IC_{50}$ of 90.65 µM against α-glucosidase, approaching the efficacy of the standard acarbose ($IC_{50}$ = 28.98 µM). These results suggest that Ag NPs could serve as promising candidates for the development of therapeutic agents targeting these enzymes. In addition to their enzyme inhibition activity, the *In Vivo* pharmacological evaluation of Ag NPs revealed potent analgesic and sedative effects. In the acetic acid-induced writhing model, Ag NPs demonstrated dose-dependent analgesic activity, achieving 83.09% inhibition at the highest dose (10 mg/kg), which is comparable to the positive control, diclofenac sodium (84.01%). The extract also exhibited analgesic activity, but it was less potent, with a maximum inhibition of 68.09% at 100 mg/kg. In the open field test, Ag NPs showed a sedative effect, as evidenced by reduced locomotor activity at higher doses. The sedative effect of Ag NPs was significant at lower doses, with 16.09% inhibition at 10 mg/kg, whereas the extract exhibited a milder sedative effect, with a maximum inhibition of 36.66% at 100 mg/kg. The molecular docking was also performed to look out the the mechanism of the interaction of the NPS with the targeted enzyme. These results are comparable with the existing literature on the green-synthesized metal NPs which shows the applicability of this work [51,52]. These findings highlight the multifaceted potential of Ag NPs synthesized using FBRE. The green synthesis approach not only provides a sustainable method for NPs production but also results in NPs with promising pharmacological activities. The ability of Ag NPs to inhibit key enzymes involved in various diseases, as well as their analgesic and sedative effects, supports their potential as therapeutic agents in treating conditions such as cancer, diabetes, and inflammation. Furthermore, their biocompatibility and

environmentally friendly synthesis route open new possibilities for their use in diagnostic and therapeutic applications, making them a valuable addition to the growing field of nanomedicine.

## 5. Conclusions

*Ficus Benghalenis* root extract (FBRE) was utilized successfully to synthesize Ag NPs where the extract was used as a bio-reductant and capping agent for a long time. The silver nitrate solution ($AgNO_3$) with *Ficus Benghalenis* root extract (FBRE) used as a precursor, biomolecules present in the plant extract are reduced the Ag NPs. These Ag NPs could be of immense use, particularly in the field of biomedical. One of its most notable SPR absorption peaks 430nm observed which is a clear induction of the perfect spherical stable NPs that have high stability and significance in the biomedical field. Their physicochemical stability, enzyme inhibitory activities, and *In Vivo* pharmacological effects demonstrate their potential in therapeutic and diagnostic fields. The use of natural products in nanotechnology holds great promise for the development of sustainable and effective biomedical materials, paving the way for future innovations in nanomedicine.

## Author contributions

**Conceptualization:** Inam Ud Din, Rahaf Ajaj, Zubair Ahmad, Naveed Muhammad, Shahid Ali, Imran Ullah.

**Data curation:** Inam Ud Din, Rahaf Ajaj, Abdur Rauf, Zubair Ahmad, Naveed Muhammad, Shahid Ali, Hassan A. Hemeg, Imran Ullah.

**Formal analysis:** Abdur Rauf, Hassan A. Hemeg.

**Funding acquisition:** Abdur Rauf.

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
