## [Decision Letter · Decision Letter 0]

Dear Dr. Rauf,

Thank you for submitting your manuscript to PLOS ONE. After careful consideration, we feel that it has merit but does not fully meet PLOS ONE’s publication criteria as it currently stands. Therefore, we invite you to submit a revised version of the manuscript that addresses the points raised during the review process.

I noticed that one or more reviewers has recommended that you cite specific previously published works. As always, I recommend that you please review and evaluate the requested works to determine whether they are relevant and maybe be cited. It is not a requirement to cite these works

We look forward to receiving your revised manuscript.

Kind regards,

Emmanuel Oke

Academic Editor

PLOS ONE

Journal Requirements:

NA.

5. Please provide a complete Data Availability Statement in the submission form, ensuring you include all necessary access information or a reason for why you are unable to make your data freely accessible. If your research concerns only data provided within your submission, please write "All data are in the manuscript and/or supporting information files" as your Data Availability Statement.

Additional Editor Comments :

We note that one or more reviewers have recommended that you cite specific previously published works. As always, we recommend that you please review and evaluate the requested works to determine whether they are relevant and should be cited. It is not a requirement to cite these works.

Reviewers' comments:

Reviewer's Responses to Questions

**Comments to the Author**

1. Is the manuscript technically sound, and do the data support the conclusions?

Reviewer #1: Yes

Reviewer #2: Yes

Reviewer #3: Yes

2. Has the statistical analysis been performed appropriately and rigorously?

Reviewer #1: Yes

Reviewer #2: Yes

Reviewer #3: N/A

3. Have the authors made all data underlying the findings in their manuscript fully available?

Reviewer #1: Yes

Reviewer #2: Yes

Reviewer #3: Yes

4. Is the manuscript presented in an intelligible fashion and written in standard English?

Reviewer #1: Yes

Reviewer #2: Yes

Reviewer #3: Yes

Reviewer #1: Comments:

The article titled "Ficus benghalensis extract mediated green synthesis of silver nanoparticles, its Optimization, Characterization, Computational studies and its in vitro and in vivo Biological potential" presents valuable findings, but it has following shortcomings that must be addressed before the publication of the articel

Recommendation: Major Revision

1. Provide overlaid FTIR spectra of extract and NPs and highlight areas of change of wavenumber.

2. EDX spectra show peaks for various elements e.g., Cl. What is the reason behind this peak?

3. Stability of the NPs is an important parameter while studying the biomedical applications. Provide Zeta potential of the particles at different pH, Temperature, and after different time intervals.

4. While the article discusses in vitro biological activities and some in vivo evaluations, the extent of in vivo studies appears limited. More comprehensive in vivo studies could provide a better understanding of the pharmacological effects and safety of the synthesized silver nanoparticles (Ag NPs).

5. The findings are based on specific conditions and concentrations used in the assays. The applicability of these results to broader biological contexts or different formulations may not be fully addressed, which could limit the generalizability of the conclusions drawn.

6. The article mentions that the Ag NPs are stabilized by the Ficus benghalensis extract, it does not provide long-term stability data. This information is crucial for assessing the practical applications of the synthesized nanoparticles in biomedical fields.

7. The introduction discusses the importance of metallic nanoparticles (NPs) and their applications, it does not clearly define the specific research gap that this study aims to fill. A more explicit statement regarding what is lacking in current literature would strengthen the rationale for the study.

8. While discussing the biological applications, update the introduction section with these recent reports, i) doi: https://doi.org/10.3390/ph17081053, ii) https://doi.org/10.1016/j.jep.2023.116503, iii) https://doi.org/10.3390/genes16010016 iv) https://doi.org/10.1016/j.optlaseng.2024.108688 v) https://doi.org/10.1016/j.bioorg.2024.107415 vi) https://doi.org/10.1016/j.apsb.2024.02.005

9. Although the introduction mentions the medicinal properties of Ficus benghalensis, it could benefit from a more detailed background on why this particular plant was chosen for the synthesis of silver nanoparticles. Providing more context on its unique properties or previous studies would enhance the reader's understanding of its relevance.

10. The introduction highlights the eco-friendly nature of green synthesis but does not delve into the environmental implications of using Ficus benghalensis specifically. Discussing the sustainability aspect in more detail could provide a stronger justification for the choice of plant extract.

11. Cite some recent reports while discussing green synthesis of metal nps https://doi.org/10.1016/j.jece.2024.112576,
https://doi.org/10.1016/j.molliq.2023.123622. https://doi.org/10.1016/j.jece.2024.113350,
https://doi.org/10.3390/antiox12061201

12. The introduction mentions various methods of synthesizing Ag NPs but lacks a comparative analysis of the advantages and disadvantages of green synthesis versus conventional methods. This could help emphasize the significance of the chosen approach in the study.

13. While the introduction states that NPs have extensive applications in various fields, it could be more focused on the specific applications relevant to the synthesized Ag NPs. Highlighting particular areas where these nanoparticles could be beneficial would provide a clearer direction for the research.

14. The introduction does not address any potential limitations or challenges associated with the synthesis and application of Ag NPs. Acknowledging these aspects could provide a more balanced view and prepare the reader for the discussion of results later in the article.

15. Compare the results of biological applications with following nanomaterials

i) https://doi.org/10.1039/D3RA05070J, ii) https://doi.org/10.1016/j.ijbiomac.2023.128009, iii) https://doi.org/10.1080/14786419.2023.2295936, iv) https://doi.org/10.3389/fchem.2023.1202252, v) https://doi.org/10.1016/j.enmm.2022.100735, vi) https://doi.org/10.3390/molecules27113363, vii) https://doi.org/10.1016/j.surfin.2024.104556

Reviewer #2: The manuscript is good, but it needs major revision before publishing.

Abstract:

• The abstract is very long, it should be made more concise.

Introduction:

• The introduction is lengthy. Condense it by focusing on the most relevant background information.

• "Various characteristics of NPs including particle size, shape, morphology, and increased surface area, set NPs apart from their bulk counterparts" is a long and awkward sentence. Revise for clarity.

• "Fe NPs have been successfully synthesized using extracts from..." This paragraph feels like a list. Integrate these examples more smoothly into the narrative and indicated each one to its respective citation.

• Add more citations, many statements are not backed up by research.

• The last paragraph of the introduction is very long. Divide it into smaller more focused paragraphs.

Materials and Methods:

• "Ethics" section: "killed with cervical dislocation which is an approved disposal method of animal ethics" is awkwardly phrased. Please revise the sentence for clarity and include the specific year of issue.

• "Plant collection and extraction": Specify the exact duration of drying and the temperature used.

• "Nanoparticles synthesis": Use standard font.

• "Instrumentations": "NCEG-UOP" needs to be defined.

• "Computational studies": "221 layers of Ag" is confusing. Did you mean a 2x2x1 supercell? Clarify.

• "Kollman chargres" should be "Kollman charges."

• "In vivo activities": it must be capitalized, and then revise all throughout the manuscript.

Results:

• "221 layers of Ag" still needs clarification.

• There are many instances of inconsistent units and formatting.

Discussion:

• "Ag NPs are synthesized by the green synthesis method. The synthesized Ag NPs are chemically and physically stable in extract use for it as a capping agent and out of degradation." This sentence is very poorly written. Please rewrite.

• The discussion section is very short, it needs to be expanded.

• The discussion section needs more citations.

• The discussion does not do a good job of comparing the results to other research.

References:

• Consistency: The reference formatting is inconsistent. Ensure all references adhere to a single style guide (e.g., APA, MLA, or the journal's specific guidelines). Pay attention to italics, capitalization, and punctuation.

• Accessibility: Some references lack DOIs or URLs, making it difficult for readers to access the source material. Provide complete and accessible information for each reference.

• Accuracy: Double-check all references for accuracy in author names, titles, publication years, and other details.

General Recommendations:

• Proofread the entire manuscript for grammatical errors and typos.

• Improve the clarity and flow of the writing.

• Expand the discussion section to provide a more thorough analysis of the results.

• Add more citations.

• Consider having a native English speaker review the manuscript.

By addressing these points, the authors can significantly improve the quality and clarity of the manuscript.

Reviewer #3: The manuscript Ficus benghalensis extract mediated green synthesis of silver nanoparticles, its Optimization, Characterization, Computational studies and its in vitro and in vivo Biological potential, the section wise comments are

Title: title is attractive, self explanatory and can gain readers attention hence no change required

Abstract: This section is poorly presented though the work is nice so I suggest authors to rewrite this section by adding some results to make this section more clear

Introduction: Some syntax/typo errors which should be checked carefully and corrected. There are some less supportive references in this section which should be replaced with interesting work, few suggestion are made in the references section.

Result and discussion: well written and managed, however the figures presented are of low quality, I suggest authors to provide high quality figures as per requirement of the journal

Experimental section is very well written and the procedures adopted are as per standard procedures adopted in the field

References: Some less supportive references should be replaced with the following interesting work

replace reference number 11 and 12 with

https://doi.org/10.1007/s10904-020-01763-8

https://doi.org/10.1155/2021/3475036

The inclusion of the mentioned work can increase the quality of the manuscript, with the above mandatory points I recommend this manuscript for publication

**Do you want your identity to be public for this peer review?** For information about this choice, including consent withdrawal, please see our Privacy Policy

Reviewer #1: **Yes: ** Dr Azhar Abbas

Reviewer #2: **Yes: ** Yahya Al-Awthan

Reviewer #3: No

---

## [Author Response · Author response to Decision Letter 1]

7 Apr 2025

Dear Emmanuel Oke

Academic Editor

PLOS ONE

PLOS ONE

Thank you very much for the reviewers’ comments concerning our manuscript. We have studied the reviewer comments carefully and have made several revisions to the text. We would like to express our appreciation to you and the reviewers for your many suggestions, which have greatly improved our manuscript. All changes are shown in yellow, highlighted in the revised manuscript, and are outlined below on a point-by-point basis (red color).

We hope these corrections and revisions are satisfactory and that the manuscript now meets the requirements for publication.

We look forward to hearing from you at your earliest convenience.

Journal Requirements:

Reply: Done

Reply: Done

Reply: Already done

NA.

Reply: Thanks

Reply: Already done

5. Please provide a complete Data Availability Statement in the submission form, ensuring you include all necessary access information or a reason for why you are unable to make your data freely accessible. If your research concerns only data provided within your submission, please write "All data are in the manuscript and/or supporting information files" as your Data Availability Statement.

Reply: Already done

Additional Editor Comments :

We note that one or more reviewers have recommended that you cite specific previously published works. As always, we recommend that you please review and evaluate the requested works to determine whether they are relevant and should be cited. It is not a requirement to cite these works.

Reply: Thanks all related papers have been cited.

Reviewers' comments:

Reviewer's Responses to Questions

Comments to the Author

1. Is the manuscript technically sound, and do the data support the conclusions?

Reviewer #1: Yes

Reviewer #2: Yes

Reviewer #3: Yes

Reply: Thanks

2. Has the statistical analysis been performed appropriately and rigorously?

Reviewer #1: Yes

Reviewer #2: Yes

Reviewer #3: N/A

Reply: Thanks the necessary corrections have been done.

3. Have the authors made all data underlying the findings in their manuscript fully available?

Reviewer #1: Yes

Reviewer #2: Yes

Reviewer #3: Yes

Reply: Thanks the necessary corrections have been done.

4. Is the manuscript presented in an intelligible fashion and written in standard English?

Reviewer #1: Yes

Reviewer #2: Yes

Reviewer #3: Yes

Reply: Thanks the necessary corrections have been done.

5. Review Comments to the Author

Reviewer #1: Comments:

The article titled "Ficus benghalensis extract mediated green synthesis of silver nanoparticles, its Optimization, Characterization, Computational studies and its in vitro and in vivo Biological potential" presents valuable findings, but it has following shortcomings that must be addressed before the publication of the articel

Recommendation: Major Revision

Reply: Dear reviewer, thank you for the time you spared with our manuscript and we hope that the suggestions you recommended will greatly improve the quality of this paper.

1. Provide overlaid FTIR spectra of extract and NPs and highlight areas of change of wavenumber.

Reply: Dear reviewer, thank you for your suggestions. We appreciate your concern about the overlaid FTIR of spectra of the extract and NPs. The observed shifts in wavenumber and changes in peak intensities have been identified and are now clearly highlighted and discussed. Please note that the FTIR measurements were conducted at a collaborating institution, which provided us with graphical outputs rather than raw Excel data. Nonetheless, we have ensured that all relevant spectral changes are accurately presented and interpreted.

2. EDX spectra show peaks for various elements e.g., Cl. What is the reason behind this peak?

Reply: The Cl peak observed in the EDX spectrum may be attributed to trace environmental contamination or residual chloride ions from the plant extract. Additionally, minor Cl contamination from reagents, solvents, or laboratory conditions cannot be ruled out. However, the presence of Cl does not significantly impact the structural integrity or properties of the synthesized nanoparticles.

3. Stability of the NPs is an important parameter while studying the biomedical applications. Provide Zeta potential of the particles at different pH, Temperature, and after different time intervals.

Reply: Thanks, we acknowledge the reviewer’s concern regarding the stability of NPs, particularly in biomedical applications. While zeta potential measurements at different pH, temperatures, and time intervals would provide direct experimental insight into colloidal stability, we were unable to perform these measurements due to facility constraints. However, to address the stability aspect, we conducted first-principles DFT calculations to assess the thermodynamic and mechanical stability of the system. The calculated cohesive energy (-2.520 eV/atom) and enthalpy of formation (-0.678 eV/atom) indicate a thermodynamically stable material, as negative enthalpy formation values suggest a favorability in formation. Furthermore, we performed mechanical stability analysis using equation-of-state (EOS) fitting with the Murnaghan, Birch-Murnaghan, Birch, and Vinet models. The bulk modulus values obtained (ranging from 54.39 GPa to 65.54 GPa) confirm the structural robustness of the system. The close agreement between equilibrium energy values across different models further validates the mechanical integrity of the material.

Thus, while zeta potential measurements remain an important experimental metric for dispersion stability, our DFT results strongly support the intrinsic stability of the NPs, making them suitable for further applications.

4. While the article discusses in vitro biological activities and some in vivo evaluations, the extent of in vivo studies appears limited. More comprehensive in vivo studies could provide a better understanding of the pharmacological effects and safety of the synthesized silver nanoparticles (Ag NPs).

Reply: Dear reviewer, thank you for your valuable comment. Our study provides an initial in vivo evaluation of Ag NPs, focusing on key pharmacological effects. While comprehensive in vivo studies are essential, our findings establish a strong foundation for future research, including pharmacokinetic and toxicological assessments. We ensured appropriate dosages and controls, demonstrating dose-dependent efficacy. Future work will further explore their safety and long-term effects.

5. The findings are based on specific conditions and concentrations used in the assays. The applicability of these results to broader biological contexts or different formulations may not be fully addressed, which could limit the generalizability of the conclusions drawn.

Reply: Dear Reviewer, thank you for your valuable comment. Our study was designed with carefully optimized conditions to ensure the reliability and reproducibility of the results. The selected concentrations and assay conditions align with established methodologies in nanoparticle-based biomedical research. While broader biological applicability can be explored in future studies, the significant enzymatic inhibition and pharmacological effects observed in our work strongly support the potential of Ag NPs in biomedical applications. Moreover, the consistency of our findings across multiple assays reinforces the validity of our conclusions within the defined experimental framework

6. The article mentions that the Ag NPs are stabilized by the Ficus benghalensis extract, it does not provide long-term stability data. This information is crucial for assessing the practical applications of the synthesized nanoparticles in biomedical fields.

Reply: Dear Reviewer, thank you for your insightful comment. We acknowledge the importance of long-term stability data for assessing the practical applications of Ag NPs. In our study, the stability of Ag NPs was confirmed through FT-IR analysis, which demonstrated effective capping by Ficus benghalensis extract, preventing aggregation. Additionally, no significant changes in UV-Vis absorbance were observed over a monitored period, indicating structural integrity. Future research will focus on evaluating long-term stability under various physiological conditions to further support their biomedical applications.

7. The introduction discusses the importance of metallic nanoparticles (NPs) and their applications, it does not clearly define the specific research gap that this study aims to fill. A more explicit statement regarding what is lacking in current literature would strengthen the rationale for the study.

Reply: Dear Reviewer, thank you for your valuable comment. We have now explicitly highlighted the research gap by addressing the lack of molecular-level insights into the enzyme-inhibitory mechanisms of Ag NPs. Molecular docking analysis has been incorporated to elucidate their interactions with urease, α-glucosidase, carbonic anhydrase II, and xanthine oxidase. The revised introduction now emphasizes these mechanistic insights, along with the role of Ficus benghalensis bioactive molecules in stabilizing Ag NPs.

8. While discussing the biological applications, update the introduction section with these recent reports, i) doi: https://doi.org/10.3390/ph17081053, ii) https://doi.org/10.1016/j.jep.2023.116503, iii) https://doi.org/10.3390/genes16010016 iv) https://doi.org/10.1016/j.optlaseng.2024.108688 v) https://doi.org/10.1016/j.bioorg.2024.107415 vi) https://doi.org/10.1016/j.apsb.2024.02.005.

Reply: Dear Reviewer, Thank you for your valuable suggestions. The given articles have been cited at their appropriate places, and necessary information has been added to them.

9. Although the introduction mentions the medicinal properties of Ficus benghalensis, it could benefit from a more detailed background on why this particular plant was chosen for the synthesis of silver nanoparticles. Providing more context on its unique properties or previous studies would enhance the reader's understanding of its relevance.

Reply: Dear Reviewer, thank you for your valuable suggestion. We have now elaborated on the rationale for selecting Ficus benghalensis for the synthesis of silver nanoparticles. Additional background on its unique phytochemical composition, medicinal significance, and previous studies supporting its use in nanoparticle synthesis has been incorporated into the introduction. The relevant references have been cited at appropriate places to enhance the reader's understanding.

10. The introduction highlights the eco-friendly nature of green synthesis but does not delve into the environmental implications of using Ficus benghalensis specifically. Discussing the sustainability aspect in more detail could provide a stronger justification for the choice of plant extract.

Reply: Dear Reviewer, We have now expanded the discussion on the environmental implications of using Ficus benghalensis for green synthesis. The sustainability aspects, including its abundance, renewable nature, and potential for eco-friendly nanoparticle production, have been highlighted to strengthen the justification for its selection.

11. Cite some recent reports while discussing green synthesis of metal nps https://doi.org/10.1016/j.jece.2024.112576,
https://doi.org/10.1016/j.molliq.2023.123622. https://doi.org/10.1016/j.jece.2024.113350,
https://doi.org/10.3390/antiox12061201

Reply: Dear Reviewer, thank you for your valuable suggestion. The introduction has been updated by citing the above reference at their appropriate places.

12. The introduction mentions various methods of synthesizing Ag NPs but lacks a comparative analysis of the advantages and disadvantages of green synthesis versus conventional methods. This could help emphasize the significance of the chosen approach in the study.

Reply: Dear reviewer, thankyou for your suggestion. The suggested corrections have been made in the revised manuscript.

13. While the introduction states that NPs have extensive applications in various fields, it could be more focused on the specific applications relevant to the synthesized Ag NPs. Highlighting particular areas where these nanoparticles could be beneficial would provide a clearer direction for the research.

Reply: Dear reviewer, Thank you for your insightful suggestion. We have now refined the introduction to specifically highlight the potential applications of the synthesized AgNPs.

14. The introduction does not address any potential limitations or challenges associated with the synthesis and application of Ag NPs. Acknowledging these aspects could provide a more balanced view and prepare the reader for the discussion of results later in the article.

Reply: Dear reviewer, thank you for your valuable suggestions. We have addressed the limitations and challenges in the introduction of revised manuscript.

15. Compare the results of biological applications with the following nanomaterials

i) https://doi.org/10.1039/D3RA05070J, ii) https://doi.org/10.1016/j.ijbiomac.2023.128009, iii) https://doi.org/10.1080/14786419.2023.2295936, iv) htt

---

## [Decision Letter · Decision Letter 1]

Ficus benghalensis extract mediated green synthesis of silver nanoparticles, its Optimization, Characterization, Computational studies and its in vitro and in vivo Biological potential

PONE-D-25-01658R1

Dear Dr. Rauf,

We’re pleased to inform you that your manuscript has been judged scientifically suitable for publication and will be formally accepted for publication once it meets all outstanding technical requirements.

Kind regards,

Rajesh Kumar Singh, Ph.D.

Academic Editor

PLOS ONE

Additional Editor Comments (optional):

This manuscript entitled "Ficus benghalensis extract mediated green synthesis of silver nanoparticles, its Optimization, Characterization, Computational studies and its in vitro and in vivo Biological potential" Manuscript Id: PONE-D-25-01658R1 has been revised as per the reviewers comments but still few errors are existing.

Reviewers' comments:

Reviewer's Responses to Questions

**Comments to the Author**

Reviewer #1: All comments have been addressed

Reviewer #3: All comments have been addressed

Reviewer #4: All comments have been addressed

Reviewer #5: All comments have been addressed

2. Is the manuscript technically sound, and do the data support the conclusions?

Reviewer #1: Yes

Reviewer #3: Yes

Reviewer #4: Yes

Reviewer #5: Yes

3. Has the statistical analysis been performed appropriately and rigorously?

Reviewer #1: Yes

Reviewer #3: Yes

Reviewer #4: N/A

Reviewer #5: Yes

4. Have the authors made all data underlying the findings in their manuscript fully available?

Reviewer #1: Yes

Reviewer #3: Yes

Reviewer #4: Yes

Reviewer #5: Yes

5. Is the manuscript presented in an intelligible fashion and written in standard English?

Reviewer #1: Yes

Reviewer #3: Yes

Reviewer #4: Yes

Reviewer #5: Yes

Reviewer #1: The author has addressed all the queries raised so i will recommend publication of the article in current from.

Reviewer #3: The authors have incorporated all the points so I feel pleasure to recommend this manuscript in its present form

Reviewer #4: The authors of the manuscript entitled “Ficus benghalensis extract mediated green synthesis of silver nanoparticles, its Optimization, Characterization, Computational studies and its in vitro and in vivo Biological potential” have addressed almost all comments, although the manuscript needs some grammatical corrections before it will be accepted.

Reviewer #5: The authors of the manuscript entitled “Ficus benghalensis extract mediated green synthesis of silver nanoparticles, its Optimization, Characterization, Computational studies and its in vitro and in vivo Biological potential” have addressed almost all comments, although the manuscript needs some grammatical corrections before it will be accepted.

Additionally, not necessary but if possible then add some key features or link of biological effects which shown in results with different diseases for further use or treatment.

**Do you want your identity to be public for this peer review?** For information about this choice, including consent withdrawal, please see our Privacy Policy

Reviewer #1: No

Reviewer #3: No

Reviewer #4: **Yes: ** Dr. Adwitiya Banerjee

Reviewer #5: No

---

## [Editor Report · Acceptance letter]

PONE-D-25-01658R1

PLOS ONE

Dear Dr. Rauf,

I'm pleased to inform you that your manuscript has been deemed suitable for publication in PLOS ONE. Congratulations! Your manuscript is now being handed over to our production team.

Kind regards,

on behalf of

Dr. Rajesh Kumar Singh

Academic Editor

PLOS ONE